# Longitudinal Nutritional Intakes in Italian Pregnant Women in Comparison with National Nutritional Guidelines

**DOI:** 10.3390/nu14091944

**Published:** 2022-05-05

**Authors:** Fabrizia Lisso, Maddalena Massari, Micaela Gentilucci, Chiara Novielli, Silvia Corti, Leonardo Nelva Stellio, Roberta Milazzo, Ersilia Troiano, Ella Schaefer, Irene Cetin, Chiara Mandò

**Affiliations:** 1Department of Biomedical and Clinical Sciences, Università degli Studi di Milano, 20157 Milan, Italy; fabrizia.lisso@asand.it (F.L.); chiara.novielli@unimi.it (C.N.); irene.cetin@unimi.it (I.C.); 2Department of Woman, Mother and Neonate, “V. Buzzi” Children Hospital, ASST Fatebenefratelli Sacco, 20154 Milan, Italy; massari.maddalena@asst-fbf-sacco.it (M.M.); silvia.corti@unimi.it (S.C.); leonardo.nelva@unimi.it (L.N.S.); roberta.milazzo@unimi.it (R.M.); 3Department of Woman, Mother and Neonate, “L. Sacco” Hospital, ASST Fatebenefratelli Sacco, 20157 Milan, Italy; micagentilucci@gmail.com; 4Nutrition and Dietetics Technical Scientific Association-ASAND, 90144 Palermo, Italy; ersilia.troiano@gmail.com; 5Bayer Consumer Care AG, 4052 Basel, Switzerland; ella.schaefer@bayer.com

**Keywords:** pregnancy, diet, nutrition, macronutrients, micronutrients, guidelines, food frequency questionnaire

## Abstract

Background: Nutritional quality during pregnancy is crucial for mother and child health and their short/long-term outcomes. The aim of this study is to evaluate the adherence to nutritional recommendations in Italy during the three pregnancy trimesters in Normal Weight (NW) and Over Weight (OW) women. Methods: Data from a multicenter randomized controlled trial included 176 women (NW = 133; OW = 43) with healthy singleton pregnancies enrolled within 13 + 6 weeks of gestation. Dietary intake was assessed every trimester by a Food Frequency Questionnaire. Results: OW and NW had similar gestational weight gain. However, as Institute of Medicine (IOM) recommend lower gestational weight gain (GWG) for OW, they exceeded the suggested range. In both groups, caloric intake during the three trimesters never met recommendations. Protein intake in first and second trimester was higher than recommendations, as was sugars percentage. Dietary fiber intake was lower in OW. Polyunsaturated fatty acids, calcium, iron and folic acid requirements were never satisfied, while sodium intake exceeded recommendations. Conclusions: NW and OW women in Italy do not adhere to nutritional recommendations during pregnancy, with lower caloric intake, protein and sugars excess and inadequacies in micronutrients intake. Pregnant women in Italy should be provided with an adequate counseling and educational intervention as well as supplementation when indicated.

## 1. Introduction

High food quality, as well as adequate macro- and micronutrient intake in pregnancy, is critical for the health of mother and child [1,2]. Nutritional requirements increase during pregnancy in order to maintain maternal metabolism and to support fetal growth and development [3,4]. Poor dietary intake or deficiencies in key micronutrients and macronutrients may have a strong impact on pregnancy outcomes and neonatal health [5], increasing the risk of several pathologies such as congenital malformations, miscarriage, preeclampsia, gestational diabetes, preterm birth and low birth weight [6,7,8,9,10]. In undernourished populations, low daily energy and protein intake in pregnant women leads to increased risk of low-birth-weight neonates and still-birth [11]. Conversely, higher energy intake from carbohydrates and fats has been associated with increased neonatal adiposity and related complications [12]. Poor diet quality in pregnancy has also been associated with higher birthweight and increased risk of large for gestational age fetuses, independent of maternal obesity and other covariates [13]. On the contrary, benefits of a high-quality diet for maternal and fetal health are well documented [4,5]. A dietary pattern such as the Mediterranean diet, characterized by the high intake of vegetables, fruits and omega-3 (n-3) polyunsatured fatty acids (PUFAs) is associated with reduced risk of several pregnancy complications [14,15,16,17,18].

Alarmingly, in developed countries, energy and nutrient intakes during pregnancy have been reported consistently inadequate compared to recommendations, with a worryingly poor micronutrient intake during all trimesters [19,20,21,22,23,24]. Interestingly, Martin et al. [22] highlighted that diet and lifestyle are the only modifiable risk factors that women can change, also underlying the importance of nutritional habits on gestational weight gain (GWG) and, consequently, on pregnancy outcomes.

For these reasons, it is crucial to evaluate the quality of the maternal diet following the updated World Health Organization (WHO) guidelines on antenatal care recommendations for a positive pregnancy experience and the International Federation of Gynecology and Obstetrics [25,26].

There is a knowledge gap concerning the adherence of nutritional habits with national guidelines in Italian pregnant women. Therefore, in the present study we carried out an analysis of dietary intake during pregnancy in Italy to evaluate the adherence to pregnancy nutritional recommendations in a population of healthy pre-gestational Normal Weight (NW) and Over Weight (OW) pregnant women using a semi-quantitative Food Frequency Questionnaire (FFQ) in order to identify any energy and nutrient intake inadequacies.

## 2. Materials and Methods

### 2.1. Study Design

This study is part of a multicenter, parallel, randomized controlled trial carried out in two Italian centers aimed at comparing the effects of once daily supplementation with multiple micronutrients and docosahexaenoic acid (DHA) versus no supplementation during pregnancy, evaluating maternal biomarkers and infant anthropometric parameters. The study was conducted from September 2016 to December 2019. Results of primary and secondary outcomes have been described elsewhere [27]. For the present analysis, we selected women with collected nutritional data in order to compare the caloric and nutritional intake of Italian pregnant women with Italian nutritional recommendations (LARN—Nutrients and Energy Reference Intake for Italian Population) [28] through the three trimesters of gestation.

Women were enrolled between 11 + 0 and 13 + 6 weeks of gestation. Six visits were carried out during the trial, from screening to final follow-up, as outlined in Figure 1. Further protocol details have been reported in [27]. This study was conducted according to the guidelines laid down in the Declaration of Helsinki, and all procedures involving human subjects/patients were approved by the Institutional Ethical Committee “Comitato Etico Milano Area 1” (reference number 11187/2016). Written informed consent was obtained from all subjects/patients. The trial was registered at ClinicalTrials.gov (Identifier: NCT04438928; url: https://clinicaltrials.gov/ct2/show/NCT04438928 accessed on 29 March 2022).

### 2.2. Study Population

Healthy, pregnant Caucasian women aged 18–42 years were screened during their first trimester prenatal visit (gestational age (GA), week 11–14) at the Obstetrics and Gynecology Units of the Sacco Hospital and the Buzzi Children Hospital in Milan, Italy.

Women were included in the study if they were having a singleton pregnancy, hemoglobin level >105 g/L, normal ultrasound examination and inconspicuous fetal anomaly screening and were taking at least 400 μg folate per day. Women were excluded if they had experienced previous adverse pregnancy outcomes, if pre-pregnancy Body Mass Index (BMI) was <18 or >30 kg/m^2^, followed a specific diet or were taking DHA/multivitamin supplements (except folate or iron). Full inclusion and exclusion criteria have been previously reported [27].

For the current evaluation, the study group was divided in two groups according to pre-pregnancy BMI, following the Institute of Medicine (IOM) guidelines [29]: Normal Weight (NW: 18–24.9 kg/m^2^) and Overweight (OW: 25–30 kg/m^2^).

FFQs were collected in the first trimester of pregnancy for subjects randomized in the supplementation study (*n* = 176; NW = 133; OW = 43). During pregnancy, 46 subjects discontinued the study, mainly because of adverse events (*n* = 32; 69.6%); for these subjects, FFQs were not collected after discontinuation. Therefore, we considered 162 subjects for the second trimester nutritional interview (NW = 122; OW = 40) and 142 subjects for the third trimester nutritional interview (NW *n* = 105; OW *n* = 37).

### 2.3. Dietary Intake Assessment

Women enrolled in this study received a standard nutritional counseling during the first visit of pregnancy. In each trimester—as described in Figure 1—they were asked to answer the FFQ we proposed, in order to register their food frequency of consumption, as well as calories and nutrients intakes.

We used a semi-quantitative FFQ consisting of 101 food items to assess the usual daily intake of foods and nutrients. The FFQ was a modified variation of a previous version (based on a Harvard questionnaire [30]), developed and validated by Vioque et al. [31]. We then referred to Italian food intake and frequency recommendations by LARN [28]. Indeed, in addition to the bromatological analyses, we organized the food items in 21 food categories in order to allow the comparison to national nutritional guidelines on food frequency consumption during pregnancy.

All participants were asked to mark their own frequencies of consumption for each item, represented by standard recommended portions by LARN. Frequencies were identified by daily consumption (6 or more, 4–5, 2–3 or 1 time per day), weekly consumption (5–6 or 2–4 times per week) and monthly consumption (1–3, 1, 1 or less, 0 times per month).

Mean daily consumption of each item and the related calories and nutrients intake were calculated. Nutritional intake was assessed in a personal interview conducted by trained medical personnel. Women were interviewed three times in order to obtain caloric and nutritional intakes for each trimester: trimester I (GA week 13–15), trimester II (GA week 24–26) and trimester III (GA week 34–36).

### 2.4. Dietary Bromatological Analysis

Bromatology provides data about the specific macro- and micronutrients content of food and beverages. Data collected by FFQ were entered in a Microsoft Excel spreadsheet (Microsoft Corporation, Redmond, WA, USA). Each frequency consumption was reported as a factor, derived from the ratio of the frequency per day, week and month each food item was consumed, in order to obtain a daily fraction of consumption for each item. Correspondences between frequencies of consumption and ratios are shown in Appendix A. Factors derived from ratios were multiplied by the bromatological composition of each food item in order to estimate the average daily intake of macro- and micronutrient. Macro- and micronutrient intakes were assessed using Food Composition Database for Epidemiological Studies in Italy (Banca Dati di Composizione degli Alimenti per Studi Epidemiologici in Italia—BDA) [32]. An example of calculation is provided in the Appendix A.

### 2.5. Energy and Protein References: Italian LARN

In order to compare energy and nutrient intake of the study population with Italian references [28], we estimated the reference levels of energy and proteins intake for the pregnant population included in the study (Energy Intake in Pregnancy: EIP and Proteins Intake in Pregnancy: PIP). Reference values were obtained starting from LARN for the non-pregnant female population aged 30–59 years (yo) (Appendix A), considering an average height of 1.65 m (close to our study population’s average height) and an average weight of 60 kg (described as standard weight for an Italian woman).

Italian guidelines [28] suggest different coefficients to correct the basal energy expenditure for daily Physical Activity Levels (PAL) in order to estimate the amount of total energy expenditure. We considered the 3 main levels described by LARN, excluding the highest PAL of 2.1, which was inconsistent with our population possible PAL. Specifically, the EIP_PAL 1.75, EIP_PAL 1.6 and EIP_PAL 1.45 are the ideal EIP corrected for the Energy Cost of Pregnancy (ECP) of each trimester and respectively referred to the three different PAL coefficients (1.45 for sedentary activity, 1.6 for moderate activity and 1.75 for elevated activity) (Table 1). We then used the average value of the 3 PALs, i.e., 1.6, as the standard reference for comparisons in each trimester of pregnancy.

In order to obtain EIP and PIP, we added up the estimated mean energy (EIF) and proteins (PIF) intake for the Italian female population to the energetic (ECP) (+69, +266 and +496 kCal for first, second and third trimesters, respectively) and proteic (PCP) (+1, +8 and +26 g for first, second and third trimesters, respectively) costs of pregnancy according to LARN. Table 1 summarizes the estimates described in this paragraph.

### 2.6. Statistical Analysis

Data processing was performed using the SAS statistical package (SAS, Cary, NC, USA) and a Microsoft Excel spreadsheet (Microsoft Corporation, Redmond, WA, USA). Statistical analyses were performed using IBM SPSS Statistics, version 27 (IBM Corp., Armonk, NY, USA).

Data are expressed as means ± standard deviation (SD) and median. Data were tested for outliers and normality by the Kolmogorov–Smirnov test.

Groups’ data were compared according to the distribution of variables taken into account (*t*-test for parametric distribution; U-Mann–Whitney test for non-parametric distribution).

The level of significance was set at *p* < 0.05.

## 3. Results

### 3.1. Subject Characteristics

Subject baseline demographics, clinical characteristics, and pregnancy characteristics for the NW and OW groups are shown in Table 2. Mean age was 31.6 ± 4.68 years. All demographics were similar between groups, with no differences, except for pre-gestational weight and pre-gestational BMI, that defined differences between groups according to IOM criteria.

### 3.2. Energy, Macro- and Micronutrients Intakes

Table 3 shows the comparison between NW and OW mean values of energy, macro- and micro-nutrients intake in each trimester.

Caloric intake decreased throughout pregnancy in OW, showing a significant difference in comparison with NW intake in the third trimester. The intake of carbohydrates, fibers and vegetable proteins showed no significant differences between groups in the first trimester, while they were significantly lower in OW in the second and third trimesters. Proteins and lipids intakes did not show significant differences between NW and OW in the first and second trimesters but were significantly lower in OW in the third trimester.

Concerning micronutrients, calcium, sodium, zinc, eicosapentaenoic acid (EPA) and DHA intake did not show significant differences between NW and OW in any trimester. We found no significant differences in potassium, phosphorous, iron and folic acid intake in the first trimester, while in both the second and third trimesters, OW showed significantly lower intakes compared to NW.

### 3.3. Comparison with Recommendations

Table 4 shows the comparison between the bromatological data of NW and OW women and the LARN Italian references established for each trimester of pregnancy, considering a mean PAL of 1.6.

Caloric requirements were not reached in any group, while protein intake, though reaching the recommended percentage (15%), was increased in the first and second trimester for NW and decreased along pregnancy for OW. Figure 2a,b shows graphical comparisons between LARN references for caloric requirement and protein intake and their respective intakes in our NW and OW populations.

Regarding the percentage of sugar intake, in each trimester, both BMI groups showed higher values than LARN recommendations, with no longitudinal significant variations. The minimum dietary fiber requirement was close to recommendations in each trimester for NW, while in OW, it was insufficient and decreased during pregnancy (Table 4). Polyunsaturated fatty acids, calcium, iron and folic acid requirements were not satisfied in either groups for each trimester (Table 4). Sodium intake was always greater than suggested (Table 4).

The caloric and nutritional data relative to the I, II and III trimesters of pregnancy are presented in Figure 3 (a, b, c, respectively).

Data are expressed as percentages of the LARN intake recommendations. The intakes were calculated from the FFQ for each nutrient.

Table 5 presents the percentage of the population that achieved portion recommendations for each food category per trimester. Portions were defined following LARN guidelines [28]. Each food category has its own references for recommended frequency, varying from one to more portions per day or sporadic/never consumption.

The calculation procedure was as follows: 21 food categories are presented, each of them composed by several food items (101 food items in total). We calculated the average frequency of consumption of each food item, referred by the pregnant patient during the interview. Then, we calculated the relative percentage of subjects that adhered to LARN recommendation in the entire population, as well as in the NW and OW subgroups.

## 4. Discussion

The aim of the present analysis was to investigate Italian pregnant women’s nutritional habits in order to compare the nutritional habits and intakes of two different pre-pregnancy BMI groups (NW and OW women) and to evaluate their adherence to Italian nutritional guidelines in pregnancy (LARN) [28].

Caloric intake during the three trimesters did not meet recommendations in both groups. Protein intake during the first and second trimesters was higher than recommendations, as well as sugar percentage. Dietary fiber intake was lower in OW. Polyunsaturated fatty acid, calcium, iron and folic acid requirements were never satisfied, while sodium intake exceeded recommendations.

GWG did not differ between the two BMI groups. However, while the GWG of NW women followed IOM recommendations (11.5–16 kg for pre-pregnancy NW BMI) [29], OW women showed higher GWG than recommended (7–11.5 kg for pre-pregnancy OW BMI). This trend might be due to the absence of a specific nutritional counseling. Indeed, if not educated, the OW individuals tend to show a low adherence to nutritional guidelines, and this can also affect pregnancy habits [33]. Although the benefits of adhering to the GWG IOM recommendations for pregnancy outcomes are well known, pregnant women face difficulties in maintaining weight gain within the suggested limits [34]. Clinicians and dietitians should improve the strategies to help pregnant women in reaching the most appropriate GWG for them [34]. Harrison et al. [35] recently reviewed clinical practice guidelines for weight management in preconception, pregnancy and postpartum published between 2010 and 2019. This review highlighted the lack of common protocols in the clinical practice worldwide for weight management. Although health professionals generally advise about desirable GWG depending on pre-pregnancy BMI, more efforts are required to improve this communication, using simple and specific advice about diet and physical activity and with periodic weight control in order to support pregnant women in reaching the best and personalized GWG.

Previous studies showed higher caloric intake in women with excessive GWG [36]. Therefore, we expected similar or higher caloric intakes in OW vs. NW women. Surprisingly, we reported a lower caloric intake in OW women compared to NW, with statistically significant differences in the second and third trimester, when the pregnancy energy demands are higher. Recently, Concina et al. [37] showed similar data, reporting lower energy intake than recommended in the Mediterranean PHIME cohort of pregnant women. The authors suggested that despite the fact that pregnant women did not achieve the energy recommendation, they did not present an energy deficit due to the combination of inadequate eating habits and low physical activity, which is particularly neglected in pregnancy and in women with elevated BMI.

Nevertheless, we found lower energy intake compared to Italian recommendations in both groups during the three trimesters of pregnancy. The same observations were previously reported by other groups [38,39,40,41] underlying the inadequacy of energy intake compared to recommendations in pregnancy, especially in the last two trimesters. A possible reason of these results is that the FFQ methodology employed in the present and in mentioned previous studies do not include a few novel foods, new ultra-processed food or ethnical cooking items that are widely consumed in the studied populations [42,43], thus possibly contributing to the increase in the real energy intake. Moreover, generally, OW patients tend to under-estimate their general food consumption. The additional employment of a detailed seven-day food recording or 24-h re-call (with weighted food records) [44] could increase the accuracy of these findings, being representative of the exact size of consumed food portions. Another hypothesis explaining the reported lower energy intakes compared to Italian LARN references is that physical activity tends to decrease along pregnancy [45], even in normally active women. Therefore, lower or personalized physical activity levels (PAL) should be used for the calculation of total energy intake during pregnancy. Nevertheless, even calculating energy intake of this study population by considering the lowest PAL (1.45), this would result lower than recommendations in all trimesters of pregnancy (as shown in Figure 2b).

### 4.1. Macronutrients and Fiber Intake

As well as for energy intake, Italian LARN recommendations suggest an increase in proteins intake during the three trimesters of pregnancy. In the present study, both NW and OW pregnant women achieved an adequate percentage of protein intake, with no significant differences among groups, as desirable. However, considering proteins intake in grams, these records were over recommendations for both groups in the first and second trimesters, reaching suggested values only in NW during the third trimester. Likely, in our study population, the sources of vegetable proteins were derived from cereals, pasta/rice and ultra-processed foods that were widely consumed in both NW and OW, rather than other vegetable proteins sources such as legumes. This is in line with Mulè et al. [46], who reported similar results in a population of Italian adults. We also observed that the percentage of women adhering to recommendations for processed and preserved meat was low in all trimesters. All things considered, although we found an adequate percentage of protein intake, the quality of proteins seemed to be poor, therefore being different from a healthy Mediterranean dietary trend, showing little benefits for metabolic health [47].

Considering carbohydrates and sugars intakes, we found two different scenarios, with the same trend in both BMI groups. Even if carbohydrates intake was lower in OW compared to NW in the second and third trimester, the comparison of both groups to LARN recommendations (45–60%) showed an adequate intake percentage for carbohydrates, contrary to a higher intake in sugars. This ranged between 20.4% and 23.8%, while it is recommended to remain below 15% per day in order to limit the impact on glycemia, insulin resistance and plasmatic lipids [28]. Concina et al. [37] confirmed the same relation between dietary habits in pregnancy and sugars intake in countries belonging to the Mediterranean PHIME cohort. Our findings can be explained by the observation that only a low percentage of women (around 30%) matched the recommendations about frequencies of consumption of the most representative categories, in terms of quality of carbohydrates (bread, pasta and rice). In contrast, we observed an excessive consumption of sweet bakery, biscuits and ultra-processed food that, together with the higher protein intake in grams, possibly drove pregnant women to a satisfying sense of satiety [48,49].

We also found a significant difference between groups in fiber intake during the second and third trimester, with a lower intake in OW women. Compared to LARN recommendations, NW women were always close to the minimum recommended along pregnancy. However, fiber intake never surpassed the recommended minimum of 25 g, as also reported by other authors in European and non-European populations [33,38,39,50,51,52]. The main markers of dietary fiber intake are fruits and vegetables. Analyzing the percentage of those women that achieved recommendations, we observed a marked decrease in fruit and vegetable intakes in OW in the course of pregnancy, possibly due to pre-pregnancy behaviors. On the contrary, a high percentage of NW matched these items well. The higher percentage of adherence to legumes frequency recommendation in NW compared to OW could contribute to daily fiber intake adequacy in this group.

The percentage of lipids suited to recommendations in both BMI groups along pregnancy. Considering lipids intake in grams, this was lower in OW than NW, with a significant difference in the third trimester. Dietary lipids composition was also lower in OW vs. NW in the third trimester. However, EPA and DHA comparisons did not reach statistical significance. Nevertheless, in the general population, despite the high percentage of women consuming ultra-processed food, saturated fatty acids percentage followed recommendations. These results differ from findings by Concina et al. [37], where total lipids and saturated fatty acids exceeded recommendations in the cohort of Italian pregnant women included in that study. On the contrary, the polyunsaturated fatty acids percentage was below the recommendations for both BMI groups, in agreement with previous findings [37]. Damen et al. [53] highlighted the importance of the correct total lipids and saturated fatty acids intake, especially during the second trimester, when fetal fat mass percentage and fetal programming have been shown to be influenced by maternal lipids intake. However, surprisingly, grams of EPA and DHA were above the minimum required. Indeed, even if the percentage of adherence to fish servings recommendations was low, over 50% of women consumed fish servings above the suggestions, ensuring optimal EPA and DHA intake (data not shown) [54,55,56].

### 4.2. Micronutrients Intake

Micronutrients intake did not follow recommendations (both in excess or deficiency), similarly to previous studies in European and Non-European populations during pregnancy [9,37].

In particular, we reported low intakes of calcium, iron and folic acid.

Calcium intake during pregnancy is important for prevention against maternal decrease in bone mineral density [57]. Our data showed that in both BMI groups and during the three trimesters, the percentage of women consuming milk and yogurt within recommendations was very low, with lower assumption in most of the women. Moreover, few women consumed the proper amount of water, thus suggesting that an improved education for the right amount and a correct choice of high-quality mineral water composition should be offered to prevent calcium deficiency in pregnant women [58].

Iron deficiency during pregnancy is among the main causes of adverse maternal and offspring outcomes and controversial effects of iron supplementation have been reported [59,60]. Our population showed low iron intake, as previously reported for the Italian population [61]. Cereals and derivatives have been previously reported as the main dietary sources of iron for Italian population [62]. Here, the percentage of women following recommendations on bread, pasta and rice consumption was low, possibly explaining the low iron dietary intake. Although low iron intake was observed in both BMI groups, OW showed a significantly lower iron intake compared to NW. As the excessive weight hinders iron bioavailability [63], these data confirm the importance of iron status and iron nutritional intake assessment starting from the beginning of pregnancy, in order to offer personalized dietary advices, according to the women’s BMI.

As well as for iron intake, the main source of folates are cereals and derivatives together with vegetables [62]. In our study, both BMI groups showed low adherence percentages to these food groups, accounting for insufficient folates intake and confirming the mandatory need of further education to encourage folate supplementation to all women that do not actively exclude pregnancy [64].

Our data confirmed that this Italian population also presented the most common micronutrients deficiencies previously reported in both the pre-conceptional and gestational period, leading to a depletion in diet quality independently from macronutrient balance, energy intake and pre-gestational BMI category [37,65,66]. This finding highlights the importance of micronutrients supplementation and the urgent need of a nutritional educational campaign to support healthy habits even in industrialized countries of the Mediterranean area [64,65]. In particular, the role of the health professional becomes more important when the patient’s educational level is medium-low, due to the direct correlation of this factor with the ability to adhere to pre-conceptional and pregnancy nutritional guidelines [66]. The effort to reach an appropriate adherence to micronutrients intake guidelines can be achieved by sharing and accepting European health policies, as well as implementing nutritional counseling and follow-ups during pregnancy. The use of technology for patients’ education and monitoring could represent a valuable aid to promote the compliance with intake recommendations [67].

Finally, as expected from food frequency evaluation, sodium intake was higher than suggested. Considering that the FFQ we proposed analyzed food intake and frequency of consumption and did not examine any added salt, it is likely that sodium intake was alarmingly even greater than what was reported. This evidence complies with the higher consumption of ultra-processed food compared to healthy lifestyle recommendations [28].

### 4.3. Strenghts and Limitations

This study longitudinally investigates nutritional habits during pregnancy, in comparison with LARN national nutritional guidelines [28]. Knowledge about the adherence to LARN in the Italian pregnant population was still lacking, and this is the first study reporting data on this topic. Moreover, one of the strengths of this manuscript is the innovative and thorough approach to the investigation of nutritional habits in pregnancy. Indeed, previous investigations mostly provided nutritional data referred to a specific trimester of pregnancy, thus lacking information about the longitudinal and dynamic change of nutritional habits and food and beverage intakes in the course of pregnancy.

However, a few limitations should also be addressed. In this study, we used a Food Frequency Questionnaire, which required reporting general nutritional behaviors relative to the whole trimester. The additional use of a 24-h recall or of a 3 or 7 days recording diary might have given a more detailed report, avoiding possible inaccuracies. Nevertheless, the FFQ allowed a quick and effortless procedure, guaranteeing an immediate deliverable. Another possible limitation of the present study is the lack of information on the women’s physical activity level during the three trimesters. Therefore, a moderate activity was generically assumed for all patients, without taking into account possible variations along the pregnancy course. Finally, differences between NW and OW in misreporting food intake or pregestational weight might have occurred. Indeed, OW women have been shown to under-report energy intake compared to women with normal weight [68].

## 5. Conclusions

The results of the present study indicate that both NW and OW women in Italy do not adhere to nutritional recommendations during pregnancy, with lower caloric and higher proteins and sugars intake. Despite the balance in macronutrient distribution, the reported data indicate that maternal nutrition in pregnancy needs an adequate survey, educational actions and personalized supplementation interventions, since a macronutrient equilibrium does not necessarily reflect good nutritional quality. Indeed, inadequacies in micronutrients intake were also evident. A detailed and dedicated counseling starting from the beginning of pregnancy, or even before, could therefore support recommendations and ameliorate macro- and micronutrients intake with positive effects on both the mother and offspring. Further efforts can be spent in the use of technology for patients’ education and monitoring. The clinical aim should be helping pregnant women in changing their own nutritional habits when incorrect, with special attention to women with abnormal pre-pregnancy BMI. Nutritional recommendations should be also evaluated in future studies in relation to health implications.

## Figures and Tables

**Figure 1 nutrients-14-01944-f001:**
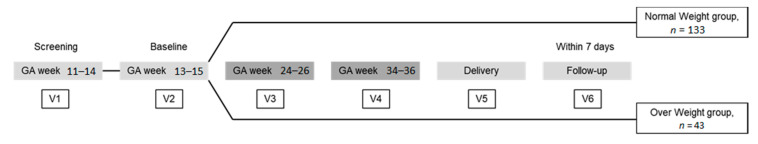
Study design. Visit 1 (V1, screening): pregnant women were screened for study eligibility. Visits 2, 3 and 4 (V2, baseline): nutritional status was assessed using a semi-quantitative FFQ. Visit 5 (V5, delivery): obstetric evaluations were performed in all women and infant anthropometric parameters were measured. Concomitant medications and adverse events were assessed at all Visits (V1–V6). GA, gestational age; FFQ, food frequency questionnaire.

**Figure 2 nutrients-14-01944-f002:**
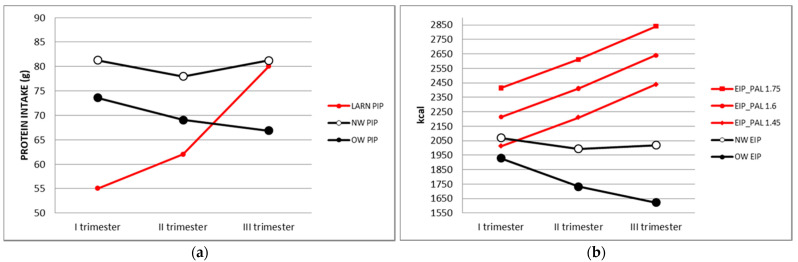
Comparison between study population energy and proteins intake through trimesters and LARN suggestions. (**a**) PIP comparison between NW, OW and LARN PIP; (**b**) EIP comparison between NW, OW and LARN EIP, considering all LARN references for PAL. LARN: Nutrients and Energy Reference Intake for Italian Population; EIP: Energy Intake in Pregnancy; PAL: Physical Activity Level; PIP: Proteins Intake in Pregnancy; NW: Normal Weight; OW: Over Weight.

**Figure 3 nutrients-14-01944-f003:**
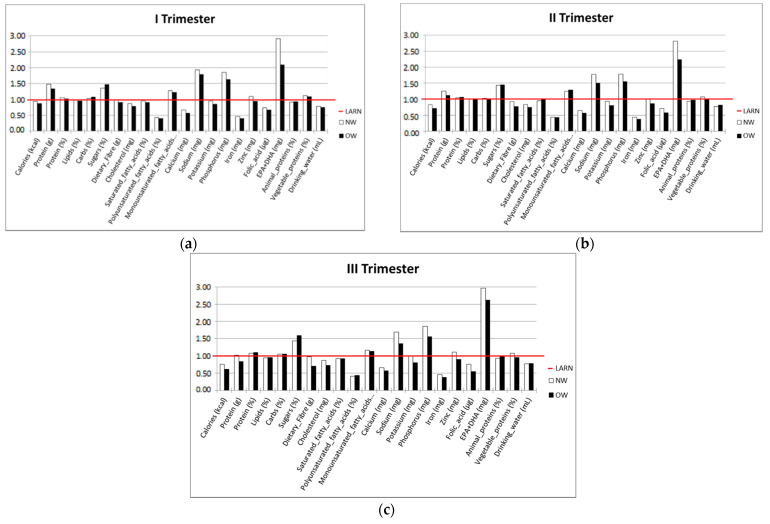
Caloric and nutritional data expressed as percentages towards LARN intake recommendations. The red line represents 100% of recommendation. Data are shown as the intakes of both subpopulations (NW and OW) for each nutrient in the three trimesters: (**a**) Percentage of energy and nutrients intakes compared to LARN recommendations for I trimester of pregnancy; (**b**) Percentage of energy and nutrients intakes compared to LARN recommendations for II trimester of pregnancy; (**c**) Percentage of energy and nutrients intakes compared to LARN recommendations for III trimester of pregnancy.

**Table 1 nutrients-14-01944-t001:** LARN energy references for Italian pregnant population aged 30–59 yo, with a mean height of 1.65 m and PALs of 1.45, 1.6 and 1.75 [28].

Female Population Aged 30–59 yo Mean Height 1.65 m	EIP I TRIM (kCal)	EIP II TRIM (kCal)	EIP III TRIM (kCal)
	ECP(69) + EIF	ECP(266) + EIF	ECP(496) + EIF
PAL 1.45	2012	2209	2439
PAL 1.6	2213	2410	2640
PAL 1.75	2414	2611	2841

LARN: Nutrients and Energy Reference Intake for Italian Population; EIP: Energy Intake in Pregnancy; ECP: Energetic Cost of Pregnancy; EIF: Energy Intake for Female; PAL: Physical Activity Level; yo: years old; TRIM, trimester.

**Table 2 nutrients-14-01944-t002:** Subject characteristics at baseline (values expressed as number of patients and mean ± standard deviation, (median)).

Characteristics	Normal Weight (NW) = 133	Over Weight (OW) = 43
Age (years)	133 31.4 ± 4.58 (32)	43 32.2 ± 5.00 (34)
Pre-gestational Weight (kg)	133 57.0 ± 6.42 (57)	43 74.1 ± 7.43 (75)
Height (m)	133 1.65 ± 0.06 (1.65)	43 1.65 ± 0.06 (1.65)
Pre-gestational body mass index (BMI) (kg/m^2^)	133 20.9 ± 1.91 (20.8)	43 27.3 ± 1.54 (26.6)
Gestational Weight Gain (GWG) (kg)	96 13.5 ± 4.59 (13)	28 14.7 ± 7.09 (13.5)
Gestational Length (weeks)	99 39.9 ± 1.07 (40)	32 39.9 ± 1.41 (40)

**Table 3 nutrients-14-01944-t003:** Energy, macronutrient and micronutrient intakes comparison between NW and. OW (values expressed as mean ± standard deviation).

Energy, Macro- and Micronutrients	I Trimester		II Trimester		III Trimester	
	NW (Mean ± st.dev)	OW (Mean ± st.dev)	NW (Mean ± st.dev)	OW (Mean ± st.dev)	NW (Mean ± st.dev)	OW (Mean ± st.dev)
Calories, kcal	2068.7 ± 759	1927.7 ± 733	1993.3 ± 761	1732.4 ± 614	2019 ± 783	1622.1 ± 623 *
Alcohol, g	0.7 ± 2	0.3 ± 1	0.9 ± 2	0.6 ± 1	1 ± 3	0.8 ± 2
Proteins, g	81.3 ± 33	73.6 ± 24	78 ± 29	69.1 ± 21	81.2 ± 35	66.8 ± 29 *
Animal proteins, g	49.8 ± 25	45.7 ± 18	48.7 ± 22	44.8 ± 16	50.6 ± 24	44.8 ± 20
Vegetable proteins, g	29.9 ± 15	26.4 ± 10	27.8 ± 11	23.1 ± 9 *	28.9 ± 13	20.9 ± 10 *
Lipids, g	68.9 ± 29	61.3 ± 22	66.2 ± 27	59 ± 21	63.5 ± 29	51.2 ± 20 *
Carbohydrates, g	284.7 ± 115	275.5 ± 132	274.8 ± 121	235 ± 102 *	283.4 ± 118	227.2 ± 104 *
Sugars, g	112.6 ± 54	112.8 ± 65	114.7 ± 62	100.2 ± 55	115.8 ± 57	103 ± 64
Dietary Fiber, g	24.6 ± 12	22.6 ± 10	23.1 ± 10	19.3 ± 8 *	24.2 ± 12	17.7 ± 10 *
Drinking water, mL	1840.2 ± 737	1747.3 ± 595	1821.1 ± 695	1916.7 ± 543	1822 ± 700	1836.1 ± 702
Cholesterol, mg	260.6 ± 124	233.3 ± 96	251.1 ± 106	225.1 ± 79	261.6 ± 125	219.1 ± 104 *
Saturated fatty acids, g	21.8 ± 9	19.5 ± 7	21 ± 9	18.9 ± 7	20.5 ± 10	16.4 ± 7 *
Polyunsaturated fatty acids, g	10 ± 5	8.6 ± 3	9.7 ± 5	8.2 ± 3	9.3 ± 5	7.8 ± 4 *
Monounsaturated fatty acids, g	29.3 ± 14	26.2 ± 11	27.6 ± 13	24.8 ± 10	26 ± 14	20.5 ± 9 *
C20_5 EPA, g	0.287 ± 0	0.205 ± 0	0.275 ± 0	0.216 ± 0	0.296 ± 0	0.253 ± 0
C22_6 DHA, g	0.44 ± 0	0.318 ± 0	0.428 ± 0	0.343 ± 0	0.447 ± 0	0.402 ± 0
Calcium, mg	798 ± 413	685.5 ± 278	769.5 ± 332	674.4 ± 292	787.8 ± 390	688.5 ± 352
Sodium, mg	2892.8 ± 1871	2677.3 ± 2015	2662.8 ± 1716	2259.8 ± 1516	2535.7 ± 1681	2029.5 ± 1174
Potassium, mg	3705.4 ± 1614	3318.9 ± 1173	3680.9 ± 1395	3153.9 ± 1092 *	3827.6 ± 1676	3125.6 ± 1712 *
Phosphorus, mg	1292.6 ± 553	1140.9 ± 352	1248.6 ± 455	1086.3 ± 360 *	1297.4 ± 548	1089.5 ± 493 *
Iron, mg	12.5 ± 8	10.7 ± 4	11.8 ± 5	10.1 ± 4 *	12.5 ± 6	10.2 ± 6 *
Zinc, mg	13.2 ± 11	11.2 ± 5	12.2 ± 6	10.3 ± 5	13.3 ± 9	10.7 ± 6
Folic acid, µg	440.8 ± 223	400.1 ± 187	427.4 ± 234	345.3 ± 180 *	455.5 ± 289	327.4 ± 217 *

st.dev: standard deviation; C20_5 EPA: eicosapentaenoic acid; C22_6 DHA: docosahexaenoic acid. U-Mann–Whitney test, * *p* < 0.05. NW vs. OW.

**Table 4 nutrients-14-01944-t004:** Comparison of NW and OW calories, macro and micronutrients intakes vs. LARN recommendations for pregnancy (values expressed as mean).

Energy, Macro and Micronutrients	BMI	I Trimester	II Trimester	III Trimester
Calories, kcal	LARN	2213	2410	2640
	NW	2068.7	1993.3	2019.0
	OW	1927.7	1732.4	1622.1
Alcohol, g	LARN	0	0	0
	NW	0.7	0.9	1.0
	OW	0.3	0.6	0.8
Proteins, g	LARN	55	62	80
	NW	81.3	78.0	81.2
	OW	73.6	69.1	66.8
Proteins, %	LARN	15	15	15
	NW	15.7	15.6	16.1
	OW	15.3	15.9	16.5
Animal proteins, %	LARN	67	67	67
	NW	61.3	62.4	62.3
	OW	62.0	64.8	67.0
Vegetable proteins, %	LARN	33	33	33
	NW	36.8	35.6	35.6
	OW	35.9	33.5	31.2
Lipids, %	LARN	20–35	20–35	20–35
	NW	30.0	29.9	28.3
	OW	28.6	30.7	28.4
Carbohydrates, %	LARN	45–60	45–60	45–60
	NW	51.6	51.7	52.6
	OW	53.6	50.9	52.5
Sugars, %	LARN	<15	<15	<15
	NW	20.4	21.6	21.5
	OW	22.0	21.7	23.8
Dietary Fiber, g	LARN	25	25	25
	NW	24.6	23.1	24.2
	OW	22.6	19.3	17.7
Drinking water, mL	LARN	2350	2350	2350
	NW	1840.2	1821.1	1822.0
	OW	1747.3	1916.7	1836.1
Cholesterol, mg	LARN	<300	<300	<300
	NW	260.6	251.1	261.6
	OW	233.3	225.1	219.1
Saturated Fatty acids, %	LARN	<10	<10	<10
	NW	9.5	9.5	9.1
	OW	9.1	9.8	9.1
Polyunsaturated Fatty acids, %	LARN	5–10	5–10	5–10
	NW	4.4	4.4	4.1
	OW	4.0	4.3	4.3
Monounsaturated Fatty acids, %	LARN	5–10	5–10	5–10
	NW	12.7	12.5	11.6
	OW	12.2	12.9	11.4
EPA + DHA, mg	LARN	350–450	350–450	350–450
	NW	727	703	743
	OW	523	559	655
Calcium, mg	LARN	1200	1200	1200
	NW	798.0	769.5	787.8
	OW	685.5	674.4	688.5
Sodium, mg	LARN	1500	1500	1500
	NW	2892.8	2662.8	2535.7
	OW	2677.3	2259.8	2029.5
Potassium, mg	LARN	3900	3900	3900
	NW	3705.4	3680.9	3827.6
	OW	3318.9	3153.9	3125.6
Phosphorus, mg	LARN	700	700	700
	NW	1292.6	1248.6	1297.4
	OW	1140.9	1086.3	1089.5
Iron mg	LARN	27	27	27
	NW	12.5	11.8	12.5
	OW	10.7	10.1	10.2
Zinc, mg	LARN	12	12	12
	NW	13.2	12.2	13.3
	OW	11.2	10.3	10.7
Folic acid, µg	LARN	600	600	600
	NW	440.8	427.4	455.5
	OW	400.1	345.3	327.4

**Table 5 nutrients-14-01944-t005:** Percentage of adherence to frequency consumption recommendations.

	Recommended Frequency	% Adherence to Recommendation I Trimester			% Adherence to Recommendation II Trimester			% Adherence to Recommendation III Trimester		
		ALL	NW	OW	ALL	NW	OW	ALL	NW	OW
Bread	2–5 portions/day	12	14	7	11	11	13	8	10	8
Breakfast Cereals	≤3 portions/week	78	75	88	77	75	83	55	75	86
Pasta/Rice	1–2 portions/day	36	36	36	31	30	33	25	32	17
Sweet bakery, biscuits	≤2 portions/week	27	26	29	26	21	38	19	23	22
Fruit	≥2 ½ portions/day	38	38	36	36	37	33	27	40	22
Vegetables	≥2 ½ portions/day	62	62	64	61	61	60	44	58	31
White meat	1–3 portions/week	43	44	43	45	45	48	31	41	47
Red meat	1–2 portions/week	42	44	33	38	38	38	29	42	42
Processed and preserved meat	≤1 portion/week	31	27	43	37	35	45	22	39	33
Fish	2–3 portions/week	18	17	19	23	23	25	12	26	25
Preserved fish	≤1 portion/week	55	55	55	54	53	55	39	60	61
Eggs	2–4 portions/week	28	30	21	28	30	23	20	28	19
Legumes	2–3 portions/week	26	29	19	23	21	30	19	25	17
Milk and Yogurt	2–3 portions/day	5	5	5	8	10	3	4	9	6
Cheese and Aged Cheese	2–3 portions/week	18	15	26	17	16	18	12	17	31
Oil	2–4 portions/day	29	29	31	26	25	30	21	25	11
Nuts	≤3 portions/week	66	65	71	69	66	83	47	65	86
Ultra-processed food	≤1 portion/month	0	0	0	1	0	0	0	0	0
Alcohols	0	71	70	76	67	66	68	50	71	69
Sugar, Honey, jam	≤2 portions/day	85	89	74	87	89	80	60	90	81
Water	≥8 portions/day	18	20	14	22	22	23	13	19	17

## Data Availability

The data that support the findings of this study are available from Bayer Consumer Care AG, but restrictions apply to the availability of these data, which were used under license for the current study, and so are not publicly available. Data are, however, available from the authors upon reasonable request and with permission of Bayer Consumer Care AG.

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
