# Peer review of "Longitudinal Nutritional Intakes in Italian Pregnant Women in Comparison with National Nutritional Guidelines"

_nutrients, 2022, doi:10.3390/nu14091944_

Round 1
Reviewer 1 Report
This longitudinal study aimed to evaluate the adherence to nutritional 18 recommendations in Italy during the three pregnancy trimesters, in Normal Weight (NW) and Over 19 Weight (OW) women.
Intro, methods: adequately written.
It would be helpful if the authors could address the following items:
Line 23 Please change sentence structure
Line 65 word change: inadequacies
Line 259-261 seems relevant for the methods?
Discussion:
- Maybe a summary para of the results at the first paragraph of discussion will be useful
- A paragraph for the strengths and limitations of the study would be useful.
Line 347-357: perhaps a discussion comparing with findings from other studies would be useful
Tables: 1. Please provide abbreviations.
- I was a bit confused as to what is EIP_PAL 1,75, EIP_PAL 1,6, EIP_PAL 1,45. Please clarify.
Author Response
We thank the reviewer for her/his suggestions that helped us to improve the manuscript. In the manuscript we highlighted modifications in red.
Please see the authors’ point-by-point responses in attachment.

Reviewer 2 Report
This study revealed the adherence to nutritional recommendations in Italy during the three pregnancy trimesters in Normal weight and Overweight. This topic is important for nutrition science. However, some critical concerns are needed to address in the manuscript. Limitations of this study do not seem to be described clearly. What is generalizability of this study?
The following are my comments.
Introduction
Is this the first study to investigate adherence to recommendations for nutrient intake among Italian pregnant women? What are results from other studies?
Results
Description of the results seems to be imprecise and unclear.
For example,
Line 177 Authors mentioned that “all demographics were similar between groups, with no differences, except for pre-gestational BMI, which defined differences between groups, according to IOM criteria.” Does pre-gestational weight in overweight women also differ from normal weight?
Line 215-
What are values of energy and nutrient intake in NW and OW in Table 4? Are these values mean energy and nutrient intake in each group?
Line 226- Authors mentioned, “The percentage of sugar intake was increased in each trimester compared to LARN recommendations.” This sentence seems to be inaccurate. Did sugar intake decrease from the first trimester to the second trimester?
Line 236- Which guideline is used for recommendations for food intake? Source of recommendations for food intake is unclear in Method section.
Table 1
The data in the Table 1 seems to be duplicated. Do both lines, the first and the third, indicate daily energy estimates based on the average value of PAL (1,6) used below as the standard reference for comparisons in each trimester of pregnancy?
Discussion and conclusion
What are the main findings of this study? What are the limitations of this study?
Line 313 “Likely, 313 in our study population the sources of vegetable proteins derived from cereals, pasta/rice and ultra-processed foods that were widely consumed in both NW and OW, rather than other vegetable proteins sources such as legumes.”
Sources of vegetable protein do not seem to be stated in the result section in this manuscript.
Author Response

(The authors gave the same response as above.)

Reviewer 3 Report
The present paper is an interestingly approaching an actual issue; it investigates the compliance of the Pregnant normal weight and overweight women in Italy, in a longitudinal food intake study on 13+6 weeks of gestation;
The authors have done a good job as they investigated the food intake compared to the national nutrition recommendations by using a validated food frequency questionnaire. The paper approaches (1) intake of energy, macronutrients (2) intake of water and micronutrients (3) detailed intake of lipids/fats (4) adherence to frequencies consumption recommendations (5) differences between pre-pregnancy normal versus overweight group (6) differences in intake between the first, second and third trimester. However, some issues must be improved.
Title
Longitudinal Nutritional Intakes in Italian Pregnant Women in Comparison with National Nutritional Guidelines
- Keywords: pregnancy; diet; nutrition; macronutrients; micronutrients; guidelines
The addition of “food frequency questionnaire” keywords would highlight more specific the paper type of research in nutrition – for further searches
- Abstract
- Please review the English language correctness and character number
- References
- Most of the references are actual.
- Please check the correctness of the format
- Introduction
- Lines 61 – did the research cover a knowledge gap?
- Materials and methods
- Line 75 – please explicitly explain the meaning of LARN
- Line 130 Bromatological analysis
- Please explain shortly what bromatological means.
- To assure the reproducibility, please provide an example of the calculation here or in the annexed documents
- Please briefly explain the PAL coefficient as is it mentioned in the referenced documents
- Statistics –
- when compared, the batch sizes are different. How does this influence the statistical results regarding the OW vs. NW pregnant womens’ nutrition?.
- Results
- Table 5 – please present the mode of calculation for adherence to frequencies consumption recommendations
- Please define the portion
- Discussions
The discussion section may follow the order of the topics approached in the results section to be easier to follow.
Limitations – please add if any
Please insert a table of abbreviations
Author Response

(The authors gave the same response as above.)

Round 2
Reviewer 2 Report
Limitation
Do authors think there could be some differences in misreporting of food and nutrient intake between NW and OW? For example, some literature has shown that women with OW are more likely to under-report energy intake than women with normal weight. Furthermore, preBMI seems to be self-reported in this study because women were recruited during pregnancy.
